behaviour, cognition, neuroscience

threat, fear, observational learning, social learning, synchrony, empathy

**Author for correspondence:**
Philip Pärnamets
e-mail: philip.parnamets@ki.se

# Physiological synchrony predicts observational threat learning in humans

Philip Pärnamets[1,2], Lisa Espinosa[1] and Andreas Olsson[1]

[1]Division of Psychology, Department of Clinical Neuroscience, Karolinska Institutet, 171 77 Stockholm, Sweden
[2]Department of Psychology, New York University, 6 Washington Place, New York, NY 10003, USA

(iD) PP, 0000-0001-8360-9097; LE, 0000-0001-7327-6815; AO, 0000-0001-5272-7744

Understanding how information about threats in the environment is shared and transmitted between individuals is crucial for explaining adaptive, survival-related behaviour in humans and other animals, and for developing treatments for phobias and other anxiety disorders. Research across species has shown that observing a conspecific's, a 'demonstrator's,' threat responses causes strong and persistent threat memories in the 'observer'. Here, we examined if physiological synchrony between demonstrator and observer can serve to predict the strength of observationally acquired conditioned responses. We measured synchrony between demonstrators' and observers' phasic electrodermal signals during learning, which directly reflects autonomic nervous system activity. Prior interpersonal synchrony predicted the strength of the observer's later skin conductance responses to threat predicting stimuli, in the absence of the demonstrator. Dynamic coupling between an observer's and a demonstrator's autonomic nervous system activity may reflect experience sharing processes facilitating the formation of observational threat associations.

In social species, like humans, knowledge about threats and dangers is often acquired through various forms of social transmission, for example, through observation. Research across species has shown that observing a conspecific's— a 'demonstrator's—threat responses to a previously neutral stimulus can cause strong and persistent threat memories in the 'observer' [1–14]. Such memories are expressed by heightened autonomic nervous system activity in the observer when later facing that stimulus alone. Observational threat learning is efficient, and minimizes risks to the individual arising from directly interacting with potential dangers [15]. Understanding how threat information in the environment is observationally acquired is central to explaining adaptive, as well as maladaptive, survival-related behaviour in humans and other animals.

Synchrony is a pervasive natural phenomenon and occurs when two systems become coupled so that their trajectories develop temporal interdependence [16,17]. It is also a fundamental feature of interpersonal coordination and social cognition [18–25]. For example, in humans, synchrony has been observed over multiple levels of analysis including intrapersonal limb coordination [17], interpersonal eye movements during communication [26], shared attention [27] and postural sway in dyadic coordination tasks [28]. Importantly, synchrony has been related to interpersonal cohesion and cooperative outcomes [21,29,30]. Related individuals will show greater synchrony in their heart rates compared with non-related individuals during hazardous social rituals [29] and dyads who show higher degrees of synchrony will show greater team cohesion [30]. Recent advances in the cognitive neurosciences have led to the discovery of coupling in BOLD fMRI signals in several contexts [19,31–33]. For example, neural coupling between individuals has been found to predict successful verbal communication, such that the more a listener's brain activity correlated with the speaker's, the better comprehension the listener would report [32]. Despite

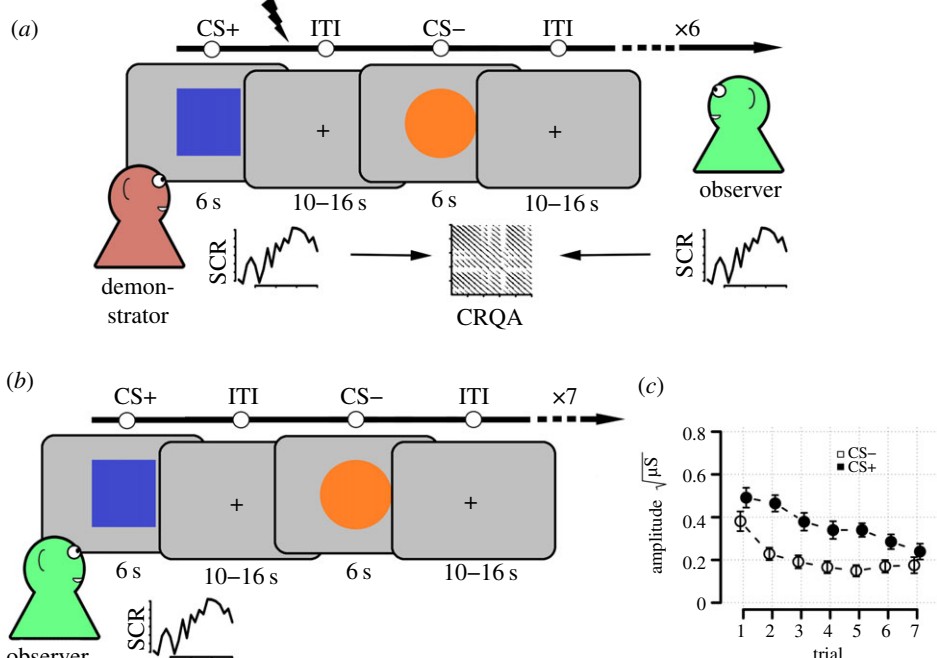

**Figure 1.** Overview of experiment. (*a*) Learning phase. Demonstrator watched two images, one that terminated with an uncomfortable electrical shock during 4 of 6 presentations (CS+) and another that never terminated with a shock (CS−). Each CS presentation lasted 6 s and inter-trial interval (ITI) varied between 10 and 16 s. Valence (CS+/−) of the first image presented was randomly varied. While the demonstrator watched the CS presentations and received shocks, the Observer watched both the demonstrator and the CS's. The Observer was instructed to learn the shock contingency. The observer received no shocks during the learning phase. Electrodermal activity was continuously recorded from both the demonstrator and observer. (*b*) Testing phase. Immediately following the learning phase, the Observer was repeatedly shown both CS's again, instructed that they would receive shocks to the same image as the demonstrator had received shocks to. Importantly, only the 7th, final presentation of the CS+ terminated with shock, to not interfere with the measurement of the vicariously acquired threat response. Greater SCR to the CS+ compared to the CS− in the Observer, in this phase, indicates successful threat learning. (*c*) Expression of learning in the testing phase. Average trial-by-trial data from testing phase showing average skin conductance responses to the CS+ (dark circles) and to the CS− (light circles) for the Observer. Error bars are 95% confidence intervals. (Online version in colour.)

considerable evidence for the role of synchrony in a diverse set of intra- and interpersonal processes, its role, if any, in social learning is not understood.

For observational learning to occur, the demonstrator's reactions to the threatening stimulus must function as unconditioned stimulus for the observer. It is possible that sharing affective states between demonstrator and observer heightens the observer's sensitivity to the demonstrator, in that way promoting learning. While this has long been hypothesized in various ways [4,11,34–37], there is no direct evidence of affective experience sharing in human threat learning. Here, we attempted to remedy this and provide evidence that synchronous patterns of arousal between observers and demonstrators influence observational threat learning.

Past experiments on observational threat learning in humans have all employed artificial situations involving confederate demonstrators, either in live settings [4–6,36,37], or more recently, displayed via video recording [9,11,13,38], and this may be one reason for the lack of evidence concerning synchrony and observational threat learning. Hence, to investigate synchrony, we adapted an existing, standard video-based paradigm [38] to a more naturalistic situation. In our paradigm, two naive participants took turns being demonstrator and observer, the demonstrator undergoing a direct conditioning procedure and the observer learning from the demonstrator's reactions (see figure 1, Methods). The experiment consisted of four blocks, each consisting of two phases, and participants switched roles halfway through (see Methods). The first phase was a learning phase where the observer watched the demonstrator receive probabilistic shocks to one of two visual images serving

as conditioned stimulus images (CS+), and never to the other (CS−). The learning phase was followed by a testing phase where both CSs were repeatedly presented again to the observer under threat of shock. To allow for the assessment of socially acquired threat responses in the absence of direct personal experience of the CS− outcome contingency, no shocks were administered to the observer following CS+ presentations during the testing phase, except following the final presentation (see Methods). Both participants' electrodermal activity was continuously recorded throughout the experiment allowing for synchrony to be calculated during the learning phase. Threat learning was measured as CS differentiation—stronger skin conductance responses to CS+ compared to CS− images in the testing phase.

# 1. Methods

## (a) Participants

We recruited a total of 138 participants who formed 69 unique demonstrator–observer dyads. Sample size was determined using simulations based on observations in an earlier pilot study (see electronic supplementary material for details). Dyads were matched by gender (24 male, 45 female). Average age was 25 years (s.d. = 4.1). Participants were recruited from the student population at Karolinska Institutet and the surrounding local community. We ensured that participants did not already know each other prior to participating in the experiment. Participants were screened from having previously partaken in conditioning experiments. Participants were given two cinema ticket vouchers as thanks for their participation. The

experimental procedure was approved by the local ethics committee (2015/2115-32).

## (b) Procedure

The experiment was divided into four blocks and followed established protocols for video-based observational learning paradigms [38]. Full details of the procedure are available in electronic supplementary material, methods.

Each block consisted of a learning phase and was followed immediately by a testing phase. In the learning phase, the observer attempted to learn the conditioned stimuli (CS) contingencies by watching the demonstrator's reactions to the CS images. The learning phase consisted of six alternating presentations of each CS+ and CS− image (see figure 1). Each CS was shown for 6 s. There was a variable length 10–16 s inter-trial interval between each CS presentation. Four of the six CS+ presentations, randomly determined, terminated with a shock to the demonstrator. Importantly, the observer received no shocks during this phase nor any instructions about which image was the CS+ (shock predicting) and which was the CS− (safe). The only way to learn this contingency for the observer was through observation.

Following the learning phase, on-screen instructions informed the observer that they would view the same two CS images again and *now* receive shocks to the *same* image that they had observed the demonstrator previously receive shocks to [9]. During this phase, the demonstrator was instructed to close their eyes and a screen was placed between demonstrator and observer occluding the observers' view of the demonstrator (electronic supplementary material, figures S1 and S2). These steps were taken to ensure that the observer would not be able to pick up any cues during the testing phase about the valence of the CS images. Hence, any expression of heightened electrodermal activity to the CS+ image compared to the CS− image would *only* reflect associations observationally formed during the previous learning phase. During the testing phase, each CS image was shown seven times. Unbeknownst to the observer, only the final CS+ presentation would terminate with a shock. This final shock was given to ensure that the observers would consider the threat of shock credible also in the next block. The whole procedure was repeated the following block, albeit with *novel* CS images ensuring no carry-over effects.

After the second block had completed, the observer was asked to rate the demonstrator on four metrics: on how much pain the demonstrator seemed to be in, how much compassion the observer felt for the demonstrator, the extent to which the demonstrator was a good model and helped them learning and to how similar to them the demonstrator appeared to be. All ratings were completed using a 16cm visual-analogue scale. The experiment then continued for two more blocks with the participants in reversed roles, meaning the participant who had been observer now became demonstrator and vice versa. Participants were unaware that this role reversal would occur. The role reversal entails that half the participants begin the experiment with two blocks as observer while the other half of the participants become observers only after having been demonstrators first for two blocks. In the results, we show that there are no effects of initial role assignment on the findings presented in this paper.

Once all four blocks had completed, the current observer rated the demonstrator on the same four metrics as introduced above. Both participant then completed an interpersonal reactivity index (IRI) [39]. Finally, participants were thanked and debriefed.

## (c) Cross-recurrence quantification analysis

To assess synchrony, we used cross-recurrence quantification analysis (CRQA), the bivariate extension of recurrence quantification analysis. Recurrence analysis is based on the analysis of recurrence plots [40]. In a recurrence plot (see figure 2), each dot marks a point of recurrence in a reconstructed phase space of the signal. The phase space is constructed using time-delay embedding. Points are considered to be recurrent if they are within some radius of one another in the resulting high dimensional phase space. Hence, three parameters need to be set to compute a recurrence plot from a time series: time delay, number of embedding dimensions and radius (see [40] for a rigorous treatment). CRQA works analogously but where the patterns of revisitation are compared between two signals [22]. CRQA yields cross-recurrence plots, analogous to regular recurrence plots.

We used the *crqa* package [41] implemented in the R statistical language to construct the cross-recurrence plots. Each cross-recurrence plot was based on the phasic skin conductance signal from the demonstrator and observer from each learning phase. The signals were down-sampled to 8 Hz and then z-scored. Optimal parameters for the CRQA analysis (delay, embedding dimensions and radius) were determined individually for each pair of signals so that they would yield an average recurrence rate between 2% and 4% [29,40]. These parameters were derived using routines from the *crqa* package and was done prior to and blind from any subsequent analyses. From each resulting cross-recurrence plot various metrics can be computed that capture the dynamics of the system being analysed [22,40,41]. Here, we computed four metrics: DETerminism, LAMinarity, maximum line (maxL) and relative Entropy (rENTR). DET represents the relative amount of recurrent points forming diagonal segments, as such DET measures the predictability of the time series as they evolve over time. LAM is analogous to DET but instead represents recurrent points forming vertical line segments, which can be thought of capturing relative stability in the system. maxL is length of the longest diagonal sequence of recurrent points, capturing the maximal strength of coupling between the two time series. rENTR calculates the Shannon entropy of the histogram of the deterministic (diagonal) sequences and indexes the complexity of the relationship between the time series.

## (d) Analysis

All analyses were performed in the R statistical language using the *brms* package [42]. We analysed the data using Bayesian multilevel regression including varying intercepts and slopes by participant and between intercept and slope correlations. All categorical regressors were deviation coded (0.5/−0.5) and all continuous regressors were standardized. Full details are given in the electronic supplementary material, Methods and regression tables for all analyses are presented in the electronic supplementary material, Results.

## 2. Results

We first evaluated the threat learning procedure and confirmed that observers readily acquired threat information from demonstrators. We tested 69 same-sex dyads and analysed observers' skin conductance responses to the onset of the CS images. We included a factor coding for if a trial consisted of a CS+ or a CS− presentation (CS). Additionally, since our experiment consisted of four blocks and participants were observers either in the first two or final two blocks of the experiment, we included variables coding for these effects in the same regression model (Block and Role, respectively). This allowed us to test if learning was equally effective for each observer regardless if this was their first or second time observing the demonstrator and if there was any difference in learning depending on the role reversal. All variables were deviation coded (see Methods).

We found a robust learned threat response indicating CS differentiation (Bayesian multilevel regression; $b = 0.15$, SE =

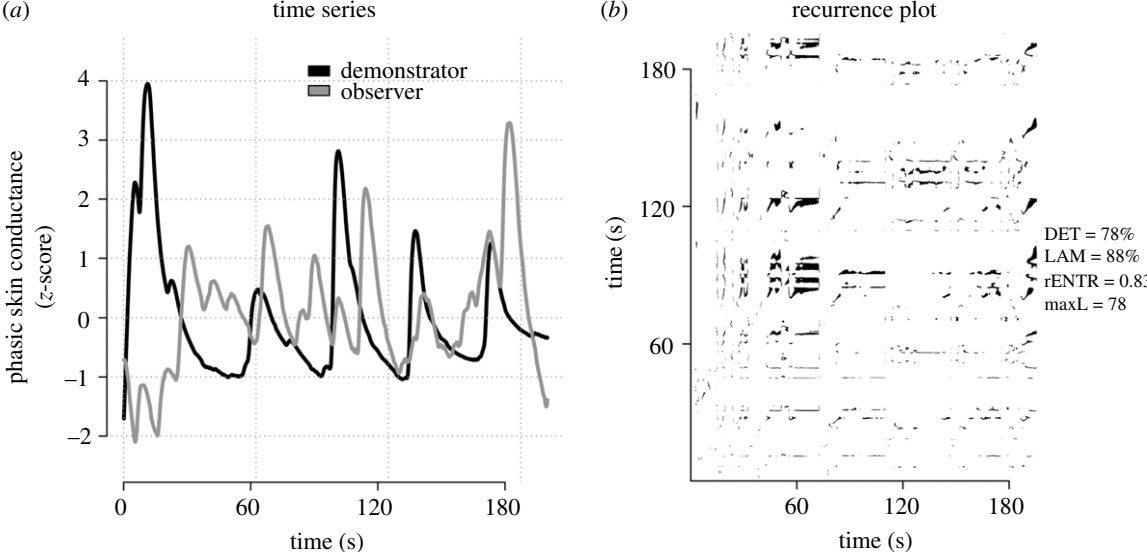

**Figure 2.** (*a*) Example of one observer's and demonstrator's phasic skin conductance time series from a single learning phase, *z*-scored to facilitate comparison. No testing phase data shown. (*b*) The resulting recurrence plot following cross-recurrence quantification analysis (CRQA) on the time series in (*a*). From each recurrence plot, four standard metrics capturing synchrony were computed; ratio of points in diagonal lines to all points (determinism, DET), maximal diagonal line length (maxL), entropy of diagonal line length distribution (rENTR) and the ratio of points in vertical lines to all points (laminarity, LAM). Resulting metrics computed from example series displayed in (*a*). Both panels (*a,b*) are displayed for illustrative purposes depicting only one representative time series and its resulting cross-recurrence plot.

0.016, CrI (95% credible interval) = [0.12, 0.19], $BF_{10}$ (Bayes factor) > $10^6$, see figure 1). Importantly, we found no interaction between CS status and Block (*b* = 0.030, SE = 0.019, CrI = [−0.007, 0.067], $BF_{10}$ = 1.34) and no interaction between CS status and Role (*b* = 0.036, SE = 0.028, CrI = [−0.018, 0.090], $BF_{10}$ = 1.26). Together, this indicates that participants learned effectively during all stages of the experiment and we conclude from this that our procedure translates standard video-based observational learning paradigms [9,38] into the more realistic situation, involving two live participants, tested here.

## (a) Synchrony predicts observational threat responses

Next, we tested the main hypothesis: that demonstrator–observer synchrony of physiological arousal during the learning phase would predict the strength of the observer's threat responses in the test phase. To quantify synchrony, we performed a cross-recurrence quantification analysis (CRQA; see figure 2, Methods). CRQA quantifies the similarity of two signals and is suitable for complex non-stationary signals where nonlinear dyanamics may exist [22,29,41,43]. We first constructed cross-recurrence plots for each dyad's phasic skin conductance time series from the learning phase portion of each block (see figure 2, for example, cross-recurrence plot). From these plots, four standard metrics of CRQA were computed that capture predictability (% DETerminism), maximum strength of coupling (maxLine) and complexity (rENTRopy) of the relationship between the time series.

We regressed each of four CRQA metrics separately on each observer's skin conductance responses to each CS image presentation from the corresponding testing phase, together with a variable indicating CS status. Interactions between the CRQA metrics and the CS variable indicate support for our hypothesis. We found strongest evidence for a link between DET and CS differentiation (*b* = 0.055, SE = 0.013, CrI = [0.029, 0.082], $BF_{10}$ = 610) and between LAM and CS differentiation (*b* = 0.051, SE = 0.014, CrI = [0.025, 0.078], $BF_{10}$ = 1481); see figure 3*a*). We also found weak evidence for maxL (*b* = 0.037, SE = 0.014, CrI = [0.010, 0.063], $BF_{10}$ = 9.7) predicting

CS differentiation, but only anecdotal evidence for rENTR (*b* = 0.027, SE = 0.012, CrI = [0.003, 0.050], $BF_{10}$ = 2.8).

While the preceding analyses indicated considerable support for our hypothesis that interpersonal synchrony during observational learning predicts later threat responses, it does not show that this effect is specific to the actual demonstrator–observer dyads from our experiment. To address this issue, we followed existing recommendations [26,30,43], and created random permutations of our data by pairing participants' across dyad boundaries. These pairings can be thought of as pseudo-dyads. The results demonstrate that the predictive effect of the CRQA metrics on threat responses does not arise between pseudo-dyads and is specific to actual demonstrator–observer pairings (see electronic supplementary material, figure 3).

## (b) Single component of CRQA metrics captures effect of synchrony on threat responses

Since the four CRQA metrics were highly correlated in our sample (*r* = 0.33 to *r* = 0.74), we followed prior work and reduced these measures into a single value using principal components analysis [30]. The factor loadings resulting from the analysis and variance explained of each factor are displayed in electronic supplementary material, table S5. The loadings suggested that the first component, which loaded roughly equally, and positively, on all metrics and captured 63% of the variance, should represent synchrony between participants best.

In line with this interpretation and our earlier analysis on the separate metrics, the first component we extracted (PC1), capturing synchrony, positively predicted CS differentiation (*b* = 0.038, SE = 0.009, CrI = [0.021, 0.056], $BF_{10}$ = 1156), see figure 3*b*). The remaining principal components did not reliably predict CS differentiation (all *b* < 0.012, all $BF_{10}$ < 0.60). In the remainder of the paper, we restrict our analyses to this synchrony component.

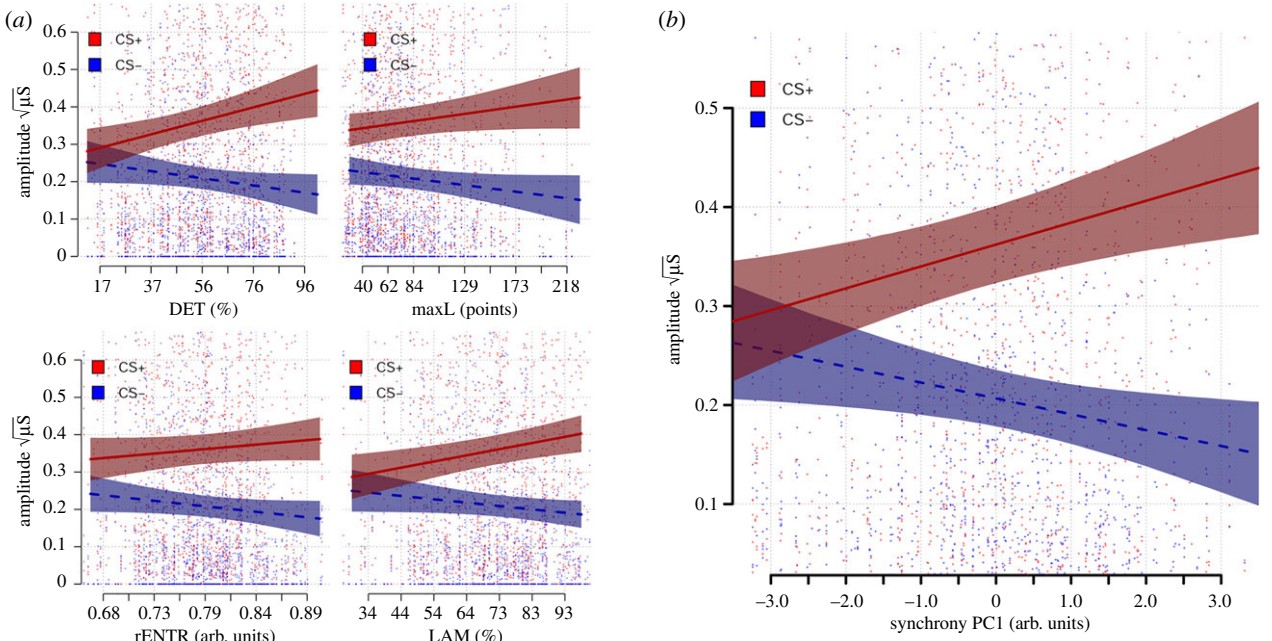

**Figure 3.** (a) Posterior predictions from regression models predicting CS+ (red, solid line) and CS– (blue, dashed line) responses as a function of each of the four metrics of synchrony computed via CRQA. (b) Posterior prediction of the observer's CS+ (red, solid line) and CS– (blue, dashed line) responses during the testing phase, as a function of synchrony during the learning phase as captured by the first principal component of the CRQA measures. Points represent individual data points from separate CS presentations. Shaded region indicates 95% posterior predictive interval. (Online version in colour.)

### (c) Stability of synchrony across trials, blocks and participant roles

Analogously to our evaluation of overall threat learning, we tested if the predictive effect of synchrony on the observer's threat responses differed depending on if the observer started the experiment in that Role or, instead, as demonstrator and if it was the first or second Block in that role. It is possible that additional experience with the experiment might in some way affect how observers and demonstrators synchronize and that our effects were only present in parts of the data. Our analyses indicated that this was not the case. We next tested if the effect of synchrony (PC1) changed over the series of consecutive CS image presentations as the participants' responses extinguished (cf. figure 1). We found that the relationship between CS differentiation and synchrony was stable across all trials in the testing phase. See electronic supplementary material, Results for details.

### (d) Specificity of synchrony as predictor of threat learning

To rule out non-synchrony based mechanisms explaining our findings, we identified three measures based on the observer's arousal in the learning phase. Each measure could plausibly capture relevant aspects of the observer's learning process. The first measure was the average strength of the observer's skin conductance response to the social unconditioned stimulus (UCS; the demonstrator receiving shocks) during the learning phase. During direct conditioning it is generally accepted that the strength of the UCS predicts the strength of later conditioned responses [44–47], and the social UCS is considered to play a similar role in observational learning [5,6]. We therefore hypothesized that the observer's reactions to the social UCS might be indicative

of its perceived strength. As such the UCS response might be capturing similar empathic processes as the synchrony component. The second measure was the average difference of the observer's responses to the CS+ over the CS– during the learning phase. While the observers were under no direct threat, it is possible that some observers began to develop responses to the CS+ anticipating the future shocks to the demonstrators. It is, therefore, possible that this early CS differentiation during the learning phase could be the source of the observed synchrony in that phase and account for the effect of synchrony we observed. Third, we included a time-lagged correlation between the demonstrators' and observers' skin conductance time series during the learning phase. Correlations capture a direct linear relationship between the signals and, unlike the previous two measures, take the whole time series into account.

The three measures outlined above exhibited low correlations with each other ($r = 0.14$ to $r = 0.20$), so we jointly regressed each of them and the synchrony component identified earlier together with their interactions with CS status on observer's skin conductance responses from the testing phase. We found that while observers' average UCS responses, during the learning phase, were positively related to the strength of their average skin conductance responses, during the testing phase, ($b = 0.045$, SE = 0.012, CrI = [0.021, 0.069], $BF_{10} = 154$), they did not interact with CS status ($b = 0.021$, SE = 0.017, CrI = [–0.012, 0.054], $BF_{10} = 0.72$). Similarly, neither learning phase CS differentiation ($b = –0.014$, SE = 0.016, CrI = [–0.044, 0.019], $BF_{10} = 0.47$) nor lagged correlations ($b = 0.007$, SE = 0.014, CrI = [–0.019, 0.033], $BF_{10} = 0.31$) interacted with CS status. Importantly, the synchrony component continued to robustly predict CS differentiation in the testing phase in this model ($b = 0.034$, SE = 0.010, CrI = [0.015, 0.054], $BF_{10} = 71$), even when accounting for these additional measures capturing other aspects of the

observer's skin conductance signals during the learning phase. Finally, we found no evidence that any of the three measures moderated the effect of synchrony on CS differentiation (all $b < 0.011$, all $BF_{10} < 0.35$).

Together these analyses show that synchrony is a specific predictor of observational threat learning and that the findings are robust to several plausible alternative predictors also derived from electrodermal activity during the learning phase.

## (e) Self-reported empathy does not account for threat learning

Next, we considered if individual differences in self-reported trait empathy (see electronic supplementary material, table S11 for descriptive statistics), as measured by the four subscales of the interpersonal reactivity index [39], predicted observational threat learning and if this could explain the effects of synchrony. We regressed the four subscales together with CS status and the synchrony component. We found no interactions between any of the subscales and CS status nor with the synchrony component (all $b < 0.01$, all $BF_{10} < 0.43$, see electronic supplementary material, table S12).

Similarly, all observers rated their perceptions of the demonstrator: how much pain the demonstrator appeared to be in, their quality as a learning model, how much compassion the observer felt for the demonstrator and how similar to the observer the demonstrator appeared to be. Again none of these measures interacted with CS status or with the synchrony component (all $b < 0.029$, all $BF_{10} < 1.43$, see electronic supplementary material, table S13).

These results suggest the momentary physiological coupling between observers and demonstrators occurs beyond participants' introspective abilities and that synchrony might constitute a more fundamental feature of empathic learning than captured by trait scales.

## 3. Discussion

We investigated if spontaneous synchrony between an observer's and a demonstrator's arousal states during observational threat learning predicted the strength of the observer's conditioned responses in a later testing phase. We found that the first principal component assessed from four common metrics of synchrony, calculated using cross-recurrence quantification analysis (CRQA), robustly predicted conditioned responses. Indeed, our findings suggested that at low levels of observer–demonstrator synchrony almost no differentiation in responses to threatening versus safe stimuli was exhibited (figure 3b). Together, our findings suggest a critical and previously undocumented facilitating role of synchrony in observational threat learning. We discuss the interpretation and implications of our findings below.

## (a) Synchrony and conditioned responses

Synchrony, as measured through CRQA, reflects similarity in the electrodermal activity trajectories of the observer and demonstrator during the learning phase. We analysed four common used metrics derived using the cross-recurrence plots from each learning phase recorded in our experiment (figure 2). These metrics capture salient patterns

in how the patterns of similarity between observers and demonstrators evolve. We found particularly strong evidence for determinism (DET) and laminarity (LAM) as predictors of later conditioned responses. Determinism implies a stronger coupling between the trajectories of the two signals, as indicated by a larger proportion of the recurrent time points form diagonal lines in the cross-recurrence plot. Laminarity suggests sustained, smooth periods in the signal's mutual evolution, as indicated by vertical segments in the cross-recurrence plots. Across all our analyses, the more synchronized demonstrators and observers were in their electrodermal activity during the observational learning phase, the stronger the observer's CS differentiation was during the testing phase.

How should synchrony be interpreted beyond the similarity of the physical properties of the two electrodermal signals? In this study, we argue that synchrony likely reflected the observer mirroring the demonstrator's autonomic nervous system trajectories as the demonstrator experienced the associations between the CS images and shocks. Consistent with previous suggestions [4,11,34–37,48], this kind of experience sharing facilitated the observer's learning of the CS–UCS contingencies even in the absence of direct experience with the shocks. The current work advances previous indirect evidence for the experience sharing hypothesis of social learning. For example, past research has found that individuals high in psychopathic traits exhibited impaired conditioned responses to a demonstrator getting shocks compared to normal controls [37] and that facial mimicry in response to watching a demonstrator getting shocks reflected experience sharing [36]. To the best of our knowledge, the results reported here represent the first direct experimental evidence that coupling of autonomic nervous system trajectories—indicative of sharing of affective sates—play a role in a learning context.

In the current study, we have a direct measure of two individual's autonomic nervous system activity. A central question for future work is how synchrony arises in the brain and how it interacts with neural systems known to be involved in threat learning. It might be possible to use hyperscanning techniques in humans using EEG or fMRI [31,49], to address this question using the experimental paradigm established here. Another promising method would be to begin with investigating if similar synchrony can be found in rodent models, where neural recordings of sub-cortical structures implicated in arousal are more readily available.

## (b) Empathic experience sharing

Behavioural and neural synchrony has been linked to mirror neurons responsible for representing the actions and intentions of social partners [18–20,50], and several accounts have attempted to extend these mechanisms to cover empathic responses [20,48,50,51]. It has long been hypothesized that sharing another person's autonomic nervous system state is the physiological substrate of empathy [1,52,53]. Some contemporary classifications of empathy consider 'experience sharing' [54] or 'empathic distress' [55] as facets of a broader empathy concept. Individual arousal levels from viewing another person in pain, as indexed by electrodermal activity, have been shown to correlate with later costly helping, which provides indirect evidence for a link between empathy and matches in arousal states [56]. Other research has reported that empathic accuracy is

greatest during periods of synchronized physiology [53] and more recently that empathic accuracy moderates synchrony in a pain task [57]. In humans, the anterior insula and the anterior cingulate cortex are part of a network responding to observational threat learning [13], and activity in these areas is known to correlate with empathy and emotion sharing in humans [55,58]. In rodents, brain regions homologous to those supporting empathy and emotion sharing in humans have been shown to be necessary for successful observational threat learning [34,35,59]. In sum, it possible, although not conclusive, that the experience sharing in our task also reflects empathic sharing of states between observers and demonstrators, although careful experimentation will be required to establish if synchrony during observational threat learning is specific to such processes or not.

Facets of empathy involving actively taking another person's perspective or reflecting mentalizing traits have also been found to facilitate observational threat learning [11,60]. For example, one study found that instructing observers to take the perspective of the demonstrator increased conditioned responses for observers who were also high in trait empathy [11]. This appraisal instruction engages mentalizing aspects of empathy that involve making explicit inferences about a partner's internal states [54,61]. In the results reported here, we found no relation between trait empathy, as captured by the interpersonal reactivity index, and conditioned responses. This suggests that we should interpret previous links between trait mentalizing empathy and observational threat learning with some caution, especially given the large sample size of the present study. There are several differences in method between the current study and studies mentioned above, most notably the absence of appraisal instructions and the use of live versus video-filmed demonstrators. In this study, trait empathy also didn't moderate the relationship between synchrony and CS differentiation. This is consistent with a recent study on empathy for pain where empathic accuracy, but not trait empathy, was a moderator for synchrony [57], even if synchrony was operationalized differently in that study compared to ours. Our findings indicate that interpersonal synchrony affects learning independently of trait empathy and this is consistent with our interpretation of synchrony reflecting experience sharing [61] or empathic distress [55].

Similarly, we found no relationship between the observer's ratings of the demonstrator, including compassion, and their conditioned responses in the testing phase. As with the trait measure, these ratings also didn't moderate the effect of synchrony. One explanation for this might be that these ratings reflect retrospective recollections of the state observers were in during the observational learning and therefore are not accurate; by contrast, synchrony will be a direct state measurement. Alternatively, interpersonal synchrony might reflect empathic processes which are more difficult to introspect on. Further work is necessary to fully understand the contributions of multiple empathic systems on processing social stimuli during threat learning.

## (c) Role of social UCS

We found that the observers' responses to the social UCS predict their general level of arousal during the testing phase, but not the strength of their later conditioned response.

This was surprising since responses to the social UCS have typically been taken to index the strength of observer's empathic responses to the demonstrator and should translate into stronger threat memories available for later recall, analogous to how a stronger UCS works during direct conditioning. In our data, observers who react strongly to seeing the demonstrator being shocked have higher response amplitudes to both the CS− and the CS+, perhaps reflecting an anxious or fear-like state during the testing phase. This suggests that the direct response to the social UCS might be less important for the observer's differential learning than previously theorized, especially when compared to mirroring and directly sharing the dynamics of the demonstrator's arousal states and might represent an uncharted difference to direct learning.

## (d) Limitations and future directions

Outstanding questions arising from this study concern the factors that can affect the degree of demonstrator–observer coupling, and how these can be manipulated, as well as the role of mediating cognitive processes. Past work has demonstrated that people tend to synchronize more with people they are more positively disposed towards [62,63] or closer to [29]. Similarly, observers learn about threats better from demonstrators who are similar to them [64]. Hence, an important avenue for future work is to manipulate the relationships between participants, for example by minimal group induction or by using natural covariates such as friendship [33], to investigate if and how this affects synchrony's role in learning. Another probably important factor is attention. It is known that observational threat learning is facilitated by increased attention to the contingencies between CS and UCS [60]. Hence, it is possible that increased attention to the demonstrator by the observer increases their interpersonal synchrony. To better understand the conditions under which observer–demonstrator synchrony emerges, it is important to examine the mediating role of attention.

Understanding the conditions when synchrony emerges is important for establishing whether synchrony affects phenomena related to observational learning, like social buffering and social safety learning [65,66]. In those paradigms observational learning protects the individual from experiencing strong threat responses and provides a safe route towards extinction of previously learned threat associations. If synchrony attunes the observer to the demonstrator's experiences then it is possible that similarly synchrony will serve to aid vicarious extinction, which in turn has clinical relevance as a model for understanding phobias and other anxiety disorders.

Data accessibility. Data, code and materials are available on the Open Science Framework (https://osf.io/mkv8c).

Authors' contributions. P.P. and A.O. designed research; P.P. and L.E. performed research; P.P. and L.E. analysed data; and P.P., L.E. and A.O. wrote the paper.

Competing interests. We declare we have no competing interest.

Funding. This research was supported by the Knut and Alice Wallenberg Foundation (KAW 495 2014.0237), a Consolidator Grant (2018-00877) from the Swedish Research Council (Vetenskapsrådet), and a Starting Grant (284366) from the European Research Council to Andreas Olsson.

Acknowledgements. We thank E. C. Nook, A. Golkar and B. Lindström for helpful comments.

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
