## [Reviewer comments · Proceedings of the Royal Society B: Biological Sciences]

Review History

RSPB-2019-2779.R0 (Original submission)

Review form: Reviewer 1

Recommendation

Accept with minor revision (please list in comments)

Scientific importance: Is the manuscript an original and important contribution to its field?

Good

General interest: Is the paper of sufficient general interest?

Good

Quality of the paper: Is the overall quality of the paper suitable?

Excellent

Is the length of the paper justified?

Yes

Should the paper be seen by a specialist statistical reviewer?

No

Do you have any concerns about statistical analyses in this paper? If so, please specify them explicitly in your report.

No

It is a condition of publication that authors make their supporting data, code and materials available - either as supplementary material or hosted in an external repository. Please rate, if applicable, the supporting data on the following criteria.

Is it accessible?

Yes

Is it clear?

Yes

Is it adequate?

Yes

Do you have any ethical concerns with this paper?

No

Comments to the Author

This paper is well-written and the main ideas are appealing and accessible to the non-specialist. However, the details included in the Results section are sometimes presented in a way that is difficult to follow. I would suggest to put them (or, at least, part of them) in a tabular form, where they can be accessed if needed, avoiding cluttering the text.

The Discussion section addresses a number of interesting issues. Since it is rather long, it would benefit having some subsections. Moreover, I miss some discussion regarding the possible impact on the conclusion that were drawn regarding the choice of parameters. For example, in pag. 9, it is said that "Optimal parameters for the CRQA analysis (...) were determined individually for each pair of signals...". Is this "tuning" potentially critical? Are there other parameters potentially sensitive to such "tuning"?

Review form: Reviewer 2

Recommendation

Major revision is needed (please make suggestions in comments)

Scientific importance: Is the manuscript an original and important contribution to its field?

Acceptable

General interest: Is the paper of sufficient general interest?

Good

Quality of the paper: Is the overall quality of the paper suitable?

Acceptable

Is the length of the paper justified?

Yes

Should the paper be seen by a specialist statistical reviewer?

No

Do you have any concerns about statistical analyses in this paper? If so, please specify them explicitly in your report.

Yes

It is a condition of publication that authors make their supporting data, code and materials available - either as supplementary material or hosted in an external repository. Please rate, if applicable, the supporting data on the following criteria.

Is it accessible?

No

Is it clear?

No

Is it adequate?

No

Do you have any ethical concerns with this paper?

No

Comments to the Author

Scientific Importance

There are a number of things to like about this paper. Most notably, the design of the study was nice, as the authors used the concept of synchrony in an interesting way to try to “capture” observational threat learning. In comparison, most research on synchrony seeks to understand more about synchrony itself. Unfortunately, the language spoke beyond the findings. If the authors kept the interpretation of results to observational threat learning and the covariance of physiological synchrony with it, then the paper would be interesting for its data alone. However, the assertions made in this paper frequently go beyond the design, data, or results.

General Interest

- The paper should be of general interest, as physiological synchrony is a construct being researched in psychology, kinesthesiology, and biomedical engineering and some researchers are examining its clinical applications.

Quality of the paper

- More evidence needs to be gathered before physiological synchrony can be considered a “biomarker” of “shared experience.” Following the classification of psychophysiological relations outlined by Cacioppo and Tassinary (1990), physiological synchrony could only be called a biomarker of shared experience if (a) it were not related to other psychological constructs (i.e., physiological synchrony has a 1:1 relationship with shared experience) and (b) the relationship between physiological synchrony and shared experience is context-dependent (i.e., it is only expected in some contexts). Without arguing about the second requirement for the term “biomarker,” the first requirement is not met: We know that physiological synchrony is related to many other psychological constructs (e.g., empathic accuracy, Levenson & Reuf, 1992; stress contagion, Waters, West, & Mendes, 2014; sociometric status, Kaplan, Burch, Bloom, & Edelberg, 1963). Covarying three potential confounding variables, as the authors did to establish specificity here, is not sufficient to establish specificity. So, it is a many-to-one relationship (i.e., viewing physiological synchrony as a physiological “outcome” of experience sharing; Cacioppo & Tassinary, 1990). This manuscript can make an equal contribution while accurately describing the relationship between physiological synchrony and shared experience.

Paper length

- Some of the details provided on processing in the main manuscript lacked sufficient details for independent evaluation (see my first comment in the statistical section), so there was no real value to having them in the main manuscript; might as well move them to the supplement and provide complete details there.

Statistical Analyses

- It would be ideal for the reader to have all the numeric information that the authors used as evidence to make any decision during data processing and analysis. For example, we learn that “SCRs were square-root transformed prior to analysis.” (p. 8), which would make sense if the data were positively skewed, which the authors had to diagnose in some way. Please provide the skewness statistic or whatever evidence was used to determine that SCRs needed to be square-root transformed. As another example, could you please provide a citation to support the specific low- and high-pass filters that you used or simply state what signals are being filtered out of the SCR waveform? An example of when these choices are supported is the citation to Haaker et al. (2017) to support the peak to peak amplitude metric of SCR. Similar to this last example, please provide the evidence (i.e., citation or statistical results) that supports all data processing and analysis decisions in the supplemental methods.

- It is important to use accepted terms when making inferences based on Bayes Factors. According to Raftery (1995), a Bayes Factor of BF_{10} would not be showing “positive” evidence for the null until it were at least as low as 0.33 and not “strong” evidence for the null (i.e., no difference) until it were .05 or lower (Jarosz & Wiley, 2014). So, the conclusion that the “observers, on average, learned equally well in their first and second blocks as observers” (p. 10) is not aligned with the inference provided from the Bayes Factor. That Bayes Factor of $BF_{10} = 1.43$ suggests that there is no evidence either way or if anything, that there is anecdotal evidence for there being a difference between the blocks (i.e., the alternative hypothesis). Please cite whatever interpretational rubric that is used for the Bayes Factors at some point and ensure that the inferential language used for all tests sticks closely to the chosen rubric (e.g., in the test that immediately follows the one given as an example on page 10 or on page 14, for the lack of a trial effect for the relationship between synchrony and CS differentiation being described as “strong” when it should just be “positive”).

- Given the high correlation among CRQA metrics (i.e., that results in them loading onto a single factor), it would be better to conduct the separate regressions on all four CRQA metrics at once as one multivariate regression. Only the multivariate statistics would need to be reported, and this would address the potential for inflated family-wise Type I error.

- Please map the CRQA metrics onto the psychophysiological constructs they are quantifying. What aspect of physiological synchrony is captured by DET and what aspect of physiology synchrony is captured by LAM, etc.?

- Minor comment: Please describe how the continuous regressors were standardized, given the hierarchical nature of the dataset. Were they standardized across all observations, within participant, within dyad, or some multistep approach?

Data

- The authors did not provide the statistical code that they used to analyze the processed data that is available on the OSF repository. Rather, they state that this code is available by request. Especially given the emphasis on quantification in the physiological synchrony literature, publicly posting the statistical syntax would help readers independently evaluate the results and would allow future researchers to more easily build on this work by using the same quantitative approach. Furthermore, sharing the code seems to be a publication requirement of the

Proceedings of the Royal Society B (i.e., on the submission form, it states, "It is a condition of publication that data, code and materials supporting your paper are made publicly available.").

- Sample size was determined using simulations based on observations in an earlier pilot study. Please provide the code on the OSF page or more information about the simulations conducted to calculate power.

- Also, please provide a Data Dictionary or Variable Coder for data_sync.txt

Decision letter (RSPB-2019-2779.R0)

21-Feb-2020

Dear Dr Pärnamets:

Your manuscript has now been peer reviewed and the reviews have been assessed by an Associate Editor. The reviewers' comments (not including confidential comments to the Editor) and the comments from the Associate Editor are included at the end of this email for your reference. As you will see, the reviewers and the Editors have raised some concerns with your manuscript and we would like to invite you to revise your manuscript to address them.

Research ethics:

Use of animals and field studies:

If you wish to submit your data to Dryad (<http://datadryad.org/>) and have not already done so you can submit your data via this link [http://datadryad.org/submit?journalID=RSPB&manu=\(Document not available\)](http://datadryad.org/submit?journalID=RSPB&manu=(Document%20not%20available)), which will take you to your unique entry in the Dryad repository.

Please submit a copy of your revised paper within three weeks. If we do not hear from you within this time your manuscript will be rejected. If you are unable to meet this deadline please let us know as soon as possible, as we may be able to grant a short extension.

Best wishes,
Dr Sasha Dall
<mailto:proceedingsb@royalsociety.org>

Associate Editor
Board Member: 1
Comments to Author:

We have now obtained two expert reviews of your manuscript, and I am happy to tell you that both the reviewers liked your paper. My own reading of your manuscript corroborates the reviews: this is an exciting piece of work that could become publishable in Proc B. However, although the reviews were generally positive, one of the reviewers raised concerns about the data processing, statistical analysis, and interpretations. This reviewer also requested that you would make the statistical code available in addition to the data. Addressing these points requires a significant revision of the text, but it should lead to an improved manuscript. The comments by the other reviewer are fewer, asking for the better organization of the result section and raising an additional point for discussion. They, too, would likely improve your paper.

Reviewer(s)' Comments to Author:

Referee: 1

Comments to the Author(s)

This paper is well-written and the main ideas are appealing and accessible to the non-specialist. However, the details included in the Results section are sometimes presented in a way that is difficult to follow. I would suggest to put them (or, at least, part of them) in a tabular form, where they can be accessed if needed, avoiding cluttering the text.

The Discussion section addresses a number of interesting issues. Since it is rather long, it would benefit having some subsections. Moreover, I miss some discussion regarding the possible impact on the conclusion that were drawn regarding the choice of parameters. For example, in pag. 9, it is said that "Optimal parameters for the CRQA analysis (...) were determined individually for each pair of signals...". Is this "tuning" potentially critical? Are there other parameters potentially sensitive to such "tuning"?

Referee: 2

Comments to the Author(s)

Scientific Importance

There are a number of things to like about this paper. Most notably, the design of the study was nice, as the authors used the concept of synchrony in an interesting way to try to "capture" observational threat learning. In comparison, most research on synchrony seeks to understand more about synchrony itself. Unfortunately, the language spoke beyond the findings. If the authors kept the interpretation of results to observational threat learning and the covariance of physiological synchrony with it, then the paper would be interesting for its data alone. However, the assertions made in this paper frequently go beyond the design, data, or results.

General Interest

- The paper should be of general interest, as physiological synchrony is a construct being research in psychology, kinesthesiology, and biomedical engineering and some researchers are examining its clinical applications.

Quality of the paper

- More evidence needs to be gathered before physiological synchrony can be considered a "biomarker" of "shared experience." Following the classification of psychophysiological relations outlined by Cacioppo and Tassinary (1990), physiological synchrony could only be called a biomarker of shared experience if (a) it were not related to other psychological constructs (i.e., physiological synchrony has a 1:1 relationship with shared experience) and (b) the relationship between physiological synchrony and shared experience is context-dependent (i.e., it is only expected in some contexts). Without arguing about the second requirement for the term

“biomarker,” the first requirement is not met: We know that physiological synchrony is related to many other psychological constructs (e.g., empathic accuracy, Levenson & Reuf, 1992; stress contagion, Waters, West, & Mendes, 2014; sociometric status, Kaplan, Burch, Bloom, & Edelberg, 1963). Covarying three potential confounding variables, as the authors did to establish specificity here, is not sufficient to establish specificity. So, it is a many-to-one relationship (i.e., viewing physiological synchrony as a physiological “outcome” of experience sharing; Cacioppo & Tassinary, 1990). This manuscript can make an equal contribution while accurately describing the relationship between physiological synchrony and shared experience.

Paper length

- Some of the details provided on processing in the main manuscript lacked sufficient details for independent evaluation (see my first comment in the statistical section), so there was no real value to having them in the main manuscript; might as well move them to the supplement and provide complete details there.

Statistical Analyses

- It would be ideal for the reader to have all the numeric information that the authors used as evidence to make any decision during data processing and analysis. For example, we learn that “SCRs were square-root transformed prior to analysis.” (p. 8), which would make sense if the data were positively skewed, which the authors had to diagnose in some way. Please provide the skewness statistic or whatever evidence was used to determine that SCRs needed to be square-root transformed. As another example, could you please provide a citation to support the specific low- and high-pass filters that you used or simply state what signals are being filtered out of the SCR waveform? An example of when these choices are supported is the citation to Haaker et al. (2017) to support the peak to peak amplitude metric of SCR. Similar to this last example, please provide the evidence (i.e., citation or statistical results) that supports all data processing and analysis decisions in the supplemental methods.

- It is important to use accepted terms when making inferences based on Bayes Factors. According to Raftery (1995), a Bayes Factor of BF_{10} would not be showing “positive” evidence for the null until it were at least as low as 0.33 and not “strong” evidence for the null (i.e., no difference) until it were .05 or lower (Jarosz & Wiley, 2014). So, the conclusion that the “observers, on average, learned equally well in their first and second blocks as observers” (p. 10) is not aligned with the inference provided from the Bayes Factor. That Bayes Factor of $BF_{10} = 1.43$ suggests that there is no evidence either way or if anything, that there is anecdotal evidence for there being a difference between the blocks (i.e., the alternative hypothesis). Please cite whatever interpretational rubric that is used for the Bayes Factors at some point and ensure that the inferential language used for all tests sticks closely to the chosen rubric (e.g., in the test that immediately follows the one given as an example on page 10 or on page 14, for the lack of a trial effect for the relationship between synchrony and CS differentiation being described as “strong” when it should just be “positive”).

- Given the high correlation among CRQA metrics (i.e., that results in them loading onto a single factor), it would be better to conduct the separate regressions on all four CRQA metrics at once as one multivariate regression. Only the multivariate statistics would need to be reported, and this would address the potential for inflated family-wise Type I error.

- Please map the CRQA metrics onto the psychophysiological constructs they are quantifying. What aspect of physiological synchrony is captured by DET and what aspect of physiology synchrony is captured by LAM, etc.?

- Minor comment: Please describe how the continuous regressors were standardized, given the hierarchical nature of the dataset. Were they standardized across all observations, within participant, within dyad, or some multistep approach?

Data

- The authors did not provide the statistical code that they used to analyze the processed data that is available on the OSF repository. Rather, they state that this code is available by request. Especially given the emphasis on quantification in the physiological synchrony literature, publicly posting the statistical syntax would help readers independently evaluate the results and would allow future researchers to more easily build on this work by using the same quantitative approach. Furthermore, sharing the code seems to be a publication requirement of the Proceedings of the Royal Society B (i.e., on the submission form, it states, "It is a condition of publication that data, code and materials supporting your paper are made publicly available.").

- Sample size was determined using simulations based on observations in an earlier pilot study. Please provide the code on the OSF page or more information about the simulations conducted to calculate power.

- Also, please provide a Data Dictionary or Variable Coder for data_sync.txt

Author's Response to Decision Letter for (RSPB-2019-2779.R0)

See Appendix A.

RSPB-2019-2779.R1 (Revision)

Review form: Reviewer 1

Recommendation

Accept as is

Scientific importance: Is the manuscript an original and important contribution to its field?

Good

General interest: Is the paper of sufficient general interest?

Good

Quality of the paper: Is the overall quality of the paper suitable?

Excellent

Is the length of the paper justified?

Yes

Should the paper be seen by a specialist statistical reviewer?

No

Do you have any concerns about statistical analyses in this paper? If so, please specify them explicitly in your report.

No

It is a condition of publication that authors make their supporting data, code and materials available - either as supplementary material or hosted in an external repository. Please rate, if applicable, the supporting data on the following criteria.

Is it accessible?

Yes

Is it clear?

Yes

Is it adequate?

Yes

Do you have any ethical concerns with this paper?

No

Comments to the Author

The authors have taken into consideration my remarks in this revised version, hence, I recommend acceptance.

Decision letter (RSPB-2019-2779.R1)

24-Apr-2020

Dear Dr Pärnamets

I am pleased to inform you that your manuscript entitled "Physiological Synchrony Predicts Observational Threat Learning in Humans" has been accepted for publication in Proceedings B.

Open Access

Paper charges

Sincerely,

Dr Sasha Dall
Editor, Proceedings B
mailto: proceedingsb@royalsociety.org

Appendix A

Responses to Reviewers' comments

Philip Pärnamets, Lisa Espinosa & Andreas Olsson

March 30, 2020

Referee #1:

Comment: *This paper is well-written and the main ideas are appealing and accessible to the non-specialist. However, the details included in the Results section are sometimes presented in a way that is difficult to follow. I would suggest to put them (or, at least, part of them) in a tabular form, where they can be accessed if needed, avoiding cluttering the text.*

Response: Thank you for this comment. In an attempt to limit the burden on reader we have moved some details of our results to the Supplementary Results giving only summaries in the main text. Additionally, we now provide regression tables for *all* our analyses in the Supplementary Results.

Comment: *The Discussion section addresses a number of interesting issues. Since it is rather long, it would benefit having some subsections.*

Response: We have added subsections to the Discussion.

Comment: *Moreover, I miss some discussion regarding the possible impact on the conclusion that were drawn regarding the choice of parameters. For example, in pag. 9, it is said that "Optimal parameters for the CRQA analysis (...) were determined individually for each pair of signals...". Is this "tuning" potentially critical? Are there other parameters potentially sensitive to such "tuning"?*

Response: Selecting non-optimal sets of parameters would likely threaten the validity of the results. The parameters are tuned maximize mutual information between the two signals and to yield recurrence rates within the specified range: 2-4% which is the recommendation from the technical literature. Due to the nature of recurrence plots, virtually any recurrence rate can be yielded by changing the analysis parameters, which would inflate the occurrence of, for example, longer segments of recurrent points which are

the basis for our further analyses (for example in the measure Determinism which capture diagonally recurring points). However, it is important to emphasize that the CRQA parameters are obtained once and blind to the later analysis we performed. Details about the technical implementation of the parameter tuning is given in Coco and Dale (2014) whose package we used for analysis.

In the revised manuscript we clarify that the parameter tuning was performed prior to and blind to later analyses.

Referee #2:

Comment: *More evidence needs to be gathered before physiological synchrony can be considered a “biomarker” of “shared experience.” Following the classification of psychophysiological relations outlined by Cacioppo and Tassinari (1990), physiological synchrony could only be called a biomarker of shared experience if (a) it were not related to other psychological constructs (i.e., physiological synchrony has a 1:1 relationship with shared experience) and (b) the relationship between physiological synchrony and shared experience is context-dependent (i.e., it is only expected in some contexts). Without arguing about the second requirement for the term “biomarker,” the first requirement is not met: We know that physiological synchrony is related to many other psychological constructs (e.g., empathic accuracy, Levenson & Reuf, 1992; stress contagion, Waters, West, & Mendes, 2014; sociometric status, Kaplan, Burch, Bloom, & Edelberg, 1963). Covarying three potential confounding variables, as the authors did to establish specificity here, is not sufficient to establish specificity. So, it is a many-to-one relationship (i.e., viewing physiological synchrony as a physiological “outcome” of experience sharing; Cacioppo & Tassinari, 1990). This manuscript can make an equal contribution while accurately describing the relationship between physiological synchrony and shared experience.*

Response: We agree with the Reviewer that the claim about "biomarker" was overstated. This claim was found in the abstract of the manuscript, and in the revised manuscript we have removed it.

Additionally, we went through the discussion to ensure that we did not make any similar overstated claims there. Our judgment is that we there provided a more measured account of our findings. Nevertheless, to guard against overinterpretation, we have added to the discussion where we discuss empathic sharing, where we now write:

In sum, it possible, although not conclusive, that the experience sharing in our task also reflects empathic sharing of states between observers and demonstrators, although careful experimentation will be required to establish if synchrony during observational threat learning is specific to such processes or not.

Comment: *Some of the details provided on processing in the main manuscript lacked sufficient details for independent evaluation (see my first comment in the statistical section), so there was no real value to having them in the main manuscript; might as well move them to the supplement and provide complete details there.*

Response: We have addressed the Reviewer’s comments below, and according to their suggestion moved most of the sections on physiological

analysis and most of the statistical analysis section to the Supplemental Methods.

Comment: *It would be ideal for the reader to have all the numeric information that the authors used as evidence to make any decision during data processing and analysis. For example, we learn that “SCRs were square-root transformed prior to analysis.” (p. 8), which would make sense if the data were positively skewed, which the authors had to diagnose in some way. Please provide the skewness statistic or whatever evidence was used to determine that SCRs needed to be square-root transformed. As another example, could you please provide a citation to support the specific low- and high-pass filters that you used or simply state what signals are being filtered out of the SCR waveform? An example of when these choices are supported is the citation to Haaker et al. (2017) to support the peak to peak amplitude metric of SCR. Similar to this last example, please provide the evidence (i.e., citation or statistical results) that supports all data processing and analysis decisions in the supplemental methods.*

Response: We apologize that our analytic decisions were not made clear in the original manuscript. Our analysis followed our standard lab protocol (cf. Olsson et al., 2007, 2016; Haaker et al., 2017), which in turn conforms to standards widely adopted in the broader fear conditioning and psychophysiological literature (cf. LaBar et al., 1995; Lykken and Venables, 1971; Boucsein, 2012).

In the revised manuscript we now write:

The raw signal from each participant was filtered offline in AcqKnowledge with a low-pass filter (1Hz) to remove potential recording artefacts and then a high-pass filter (0.05Hz) to recover the phasic skin conductance responses by removing the tonic component of the signal (Boucsein, 2012). Using CS onset and shock delivery as event markers and following established protocols (Haaker et al., 2017), skin conductance responses (SCRs) were measured as the largest peak-to-peak amplitude difference in the phasic skin conductance signal in the 0.5 to 4.5 second window following stimulus onset. Responses below $0.02\mu S$ were scored as zero. Scoring was first done using AcqKnowledge’s automated scoring algorithm, and then manually checked by an experimenter. SCRs were square-root transformed prior to analysis (LaBar et al., 1995).

Additionally, to ensure that our main findings are robust to transformations of the main outcome variables, we performed two additional analyses. We used a hurdle lognormal model (instead of a Gaussian) to model the

untransformed skin conductance responses. We replicated the first analysis for learning effects including the moderating effects of Role and Block. This is reported as Supplementary Table 2 and reproduced below as Table 1. We also replicated the analysis using the principle components of the CRQA metrics, reported as Supplementary Table 7 and reproduced below. If anything this analysis showed stronger model evidence (by a factor of 70!) for an effect of the first principle component on CS differentiation. We thank the Reviewer for providing us this opportunity to showcase the robustness of our findings.

Table 1: Results from hurdle lognormal model investigating CS differentiation and potential moderators of Role and Block. Coefficients from main model on log scale. Role, CS status and Block modeled in hurdle component, prefixed *hu*. Hurdle coefficients on logit scale.

	Estimate	Est.Error	Q2.5	Q97.5	BF_{01}	BF_{10}
intercept	-3.1801	0.1079	-3.3946	-2.9709		
Role	-0.1145	0.1979	-0.5037	0.2743	2.0919	0.4780
CS	0.8993	0.1008	0.7016	1.0986	0.0000	$> 10^6$
Block	-0.2047	0.0814	-0.3629	-0.0441	0.2688	3.7203
Role:CS	0.1442	0.1887	-0.2281	0.5160	1.9931	0.5017
Role:Block	-0.2464	0.1551	-0.5535	0.0568	0.9024	1.1081
CS:Block	0.2503	0.1436	-0.0321	0.5284	0.7612	1.3136
Role:CS:Block	-0.0920	0.2539	-0.5935	0.4111	1.8327	0.5456
hu intercept	-1.5143	0.0990	-1.7135	-1.3219		
hu Role	0.0300	0.1375	-0.2383	0.3010	1.4867	0.6726
hu CS	-0.8008	0.1131	-1.0264	-0.5842	-0.0000	$> 10^6$
hu Block	-0.0120	0.0907	-0.1867	0.1677	2.2527	0.4439

Table 2: Results from regression hurdle lognormal model using principal components of CRQA metrics. Coefficients from lognormal component on log scale. RCS status modeled in hurdle component, prefixed *hu*. Hurdle coefficients on logit scale.

	Estimate	Est.Error	Q2.5	Q97.5	BF_{01}	BF_{10}
intercept	-3.1714	0.1084	-3.3828	-2.9569		
CS	0.8941	0.0962	0.7086	1.0868	0.0000	$> 10^6$
PC1	-0.0164	0.0469	-0.1105	0.0743	10.3555	0.0966
PC2	-0.0445	0.0631	-0.1694	0.0787	6.1885	0.1616
PC3	-0.1622	0.1041	-0.3661	0.0432	1.5067	0.6637
PC4	-0.0543	0.1578	-0.3640	0.2571	3.0311	0.3299
CS:PC1	0.2417	0.0593	0.1252	0.3575	0.0000	71647.2706
CS:PC2	0.0807	0.1050	-0.1213	0.2882	3.6482	0.2741
CS:PC3	-0.0995	0.1335	-0.3630	0.1592	2.8221	0.3543
CS:PC4	0.0224	0.2006	-0.3788	0.4168	2.5501	0.3921
hu intercept	-1.4928	0.0966	-1.6884	-1.3102		
hu CS	-0.7951	0.1116	-1.0213	-0.5792	0.0000	$> 10^6$

Comment: *It is important to use accepted terms when making inferences based on Bayes Factors. According to Raftery (1995), a Bayes Factor of BF_{10} would not be showing “positive” evidence for the null until it were at least as low as 0.33 and not “strong” evidence for the null (i.e., no difference) until it were .05 or lower (Jarosz & Wiley, 2014). So, the conclusion that the “observers, on average, learned equally well in their first and second blocks as observers” (p. 10) is not aligned with the inference provided from the Bayes Factor. That Bayes Factor of $BF_{10} = 1.43$ suggests that there is no evidence either way or if anything, that there is anecdotal evidence for there being a difference between the blocks (i.e., the alternative hypothesis). Please cite whatever interpretational rubric that is used for the Bayes Factors at some point and ensure that the inferential language used for all tests sticks closely to the chosen rubric (e.g., in the test that immediately follows the one given as an example on page 10 or on page 14, for the lack of a trial effect for the relationship between synchrony and CS differentiation being described as “strong” when it should just be “positive”).*

Response: This is an important point, we have gone through the manuscript to ensure that we do not overstate evidence and to carefully point out where Bayes Factors do not give any evidence (BF: 0.3 - 3) against or for an effect. This improves the statistical reporting and we are thankful to the Reviewer for raising this concern.

Finally, we agree with the importance of giving an interpretational rubric, however, we did provide one in the original manuscript. In the Supplemental Methods, in the section concerning analysis we wrote:

We rely on Bayes Factors to make inferences about effects. We interpret Bayes Factors above 10 to constitute strong evidence for an effect, and Bayes Factors between 3 and 10 to constitute weak evidence for an effect.

We remain committed to this original rubric, which corresponds to that of Jeffreys (1961), but other readers may of course make other interpretations based on the numerical information given. We believe this is one of the strengths of the Bayesian approach generally.

Comment: *Given the high correlation among CRQA metrics (i.e., that results in them loading onto a single factor), it would be better to conduct the separate regressions on all four CRQA metrics at once as one multivariate regression. Only the multivariate statistics would need to be reported, and this would address the potential for inflated family-wise Type I error.*

Response: We agree with the Reviewer’s sentiment, however, the reason for conducting the principle component regression was precisely to deal with the high correlations between CRQA metrics. It is well known that regressing multiple correlated variables can cause masking. Indeed, following the Reviewer’s suggestion ¹ we ran a regression model testing the interaction between CS status and all of the CRQA metrics (see Table). As could be expected, the posterior estimates of all four relationships widen and with them the evidence for an effect becomes anecdotal or indeed negative. We believe this further supports our original analysis choice of using principle components regression to investigate our effects, as is supported by prior literature (Mønster et al., 2016). In the revised manuscript we report Table in the Supplemental Results.

Table 3: Results from regressing all CRQA metrics together.

	Estimate	Est.Error	Q2.5	Q97.5	BF_{01}	BF_{10}
intercept	0.286	0.016	0.255	0.316		
CS.s	0.153	0.016	0.121	0.186	0.000	$> 10^6$
DET.s	0.011	0.015	-0.019	0.040	2.614	0.382
maxL.s	-0.007	0.010	-0.027	0.014	4.037	0.248
rENTR.s	-0.008	0.009	-0.026	0.010	3.898	0.257
LAM.s	0.008	0.012	-0.015	0.032	3.106	0.322
CS.s:DET.s	0.032	0.021	-0.011	0.073	0.784	1.275
CS.s:maxL.s	0.009	0.017	-0.024	0.041	2.730	0.366
CS.s:rENTR.s	0.005	0.015	-0.024	0.035	3.249	0.308
CS.s:LAM.s	0.026	0.018	-0.008	0.061	0.891	1.123

¹We believe the Reviewer is suggesting multiple regression (regressing all CRQA metrics together), rather than multivariate regression as we only have one outcome variable.

Comment: *Please map the CRQA metrics onto the psychophysiological constructs they are quantifying. What aspect of physiological synchrony is captured by DET and what aspect of physiology synchrony is captured by LAM, etc.?*

Response: We were puzzled by this comment, since we believe we did produce such a mapping in the original manuscript. In the methods section we gave a technical introduction to all metrics:

From each resulting cross-recurrence plot various metrics can be computed that capture the dynamics of the system being analyzed (Marwan et al., 2007; Shockley, 2005; Coco and Dale, 2014). Here we computed four metrics: DETerminism, LAM-inarity, maximum line (maxL) and relative Entropy (rENTR). DET represents the relative amount of recurrent points forming diagonal segments, as such DET measures the predictability of the time-series as they evolve over time. LAM is analogous to DET but instead represents recurrent points forming vertical line segments, which can be thought of capturing relative stability in the system. maxL is length of the longest diagonal sequence of recurrent points, capturing the maximal strength of coupling between the two time series. rENTR calculates the Shannon entropy of the histogram of the deterministic (diagonal) sequences and indexes the complexity of the relationship between the time series.

In the discussion we additionally wrote:

Synchrony, as measured through CRQA, reflects similarity in the electrodermal activity trajectories of the observer and demonstrator during the learning phase. We analyzed four common used metrics derived using the cross-recurrence plots from each learning phase recorded in our experiment (see Fig 1). These metrics capture salient patterns in how the patterns of similarity between observers and demonstrators evolve. We found particularly strong evidence for determinism (DET) and laminarity (LAM) as predictors of later conditioned responses. Determinism implies a stronger coupling between the trajectories of the two signals, as indicated by a larger proportion of the recurrent time points form diagonal lines in the cross-recurrence plot. Laminarity suggests sustained, smooth periods in the signal’s mutual evolution, as indicated by vertical segments in the cross-recurrence plots. Across all our analyses, the more synchronized demonstrators and observers were in their electrodermal activity during the

observational learning phase, the stronger the observer’s CS differentiation was during the testing phase.

Comment: *Minor comment: Please describe how the continuous regressors were standardized, given the hierarchical nature of the dataset. Were they standardized across all observations, within participant, within dyad, or some multistep approach?*

Response: All continuous regressors were standardized across all observations, so that 0 always reflects the population average as does the intercept of all the statistical models and, consequently, that our population level estimates ("fixed-effects") are deviations from this average. We now clarify this in the methods section on analysis, in the Supplemental Materials where we state the procedure for standardizing variables.

Comment: *The authors did not provide the statistical code that they used to analyze the processed data that is available on the OSF repository. Rather, they state that this code is available by request. Especially given the emphasis on quantification in the physiological synchrony literature, publicly posting the statistical syntax would help readers independently evaluate the results and would allow future researchers to more easily build on this work by using the same quantitative approach. Furthermore, sharing the code seems to be a publication requirement of the Proceedings of the Royal Society B (i.e., on the submission form, it states, "It is a condition of publication that data, code and materials supporting your paper are made publicly available.").*

Response: We absolutely agree with the Reviewer here and it was an omission on our part not having done so to begin with. We have now uploaded code to the OSF repository.

Comment: *Sample size was determined using simulations based on observations in an earlier pilot study. Please provide the code on the OSF page or more information about the simulations conducted to calculate power.*

Response: We have now shared code on the OSF repository, and provide additional details in the Supplementary Methods as well, where we write:

To determine sample size we simulated data. We targeted an effect size of 0.04 (in $\sqrt{\mu S}$) for the target interaction between a CRQA metric and CS status, with a standard deviation of 0.015 for the per participant varying coefficients. Model priors were same as for our subsequent analyses. We analyzed 400 simulated datasets and assessed if $BF_{10} > 3$. Our simulations indicated that 65 dyads would provide 90% power to assess an effect.

Comment: Also, please provide a Data Dictionary or Variable Coder for `data_sync.txt`

Response: We have uploaded such a Data Dictionary to the OSF repository.

References

- W. Boucsein. *Electrodermal activity*. Springer Science & Business Media, 2012.
- M. I. Coco and R. Dale. Cross-recurrence quantification analysis of categorical and continuous time series: an r package. *Frontiers in psychology*, 5: 510, 2014.
- J. Haaker, A. Golkar, I. Selbing, and A. Olsson. Assessment of social transmission of threats in humans using observational fear conditioning. *nature protocols*, 12(7):1378, 2017.
- H. Jeffreys. *Theory of probability*, 3rd edn oxford: Oxford university press. 1961.
- K. S. LaBar, J. E. LeDoux, D. D. Spencer, and E. A. Phelps. Impaired fear conditioning following unilateral temporal lobectomy in humans. *Journal of neuroscience*, 15(10):6846–6855, 1995.
- D. T. Lykken and P. H. Venable. Direct measurement of skin conductance: A proposal for standardization. *Psychophysiology*, 8(5):656–672, 1971.
- N. Marwan, M. C. Romano, M. Thiel, and J. Kurths. Recurrence plots for the analysis of complex systems. *Physics reports*, 438(5-6):237–329, 2007.
- D. Mønster, D. D. Håkansson, J. K. Eskildsen, and S. Wallot. Physiological evidence of interpersonal dynamics in a cooperative production task. *Physiology & behavior*, 156:24–34, 2016.
- A. Olsson, K. I. Nearing, and E. A. Phelps. Learning fears by observing others: the neural systems of social fear transmission. *Social cognitive and affective neuroscience*, 2(1):3–11, 2007.
- A. Olsson, K. McMahon, G. Papenberg, J. Zaki, N. Bolger, and K. N. Ochsner. Vicarious fear learning depends on empathic appraisals and trait empathy. *Psychological science*, 27(1):25–33, 2016.
- K. Shockley. Cross recurrence quantification of interpersonal postural activity. In M. A. Riley and G. C. V. Orden, editors, *Tutorials in contemporary nonlinear methods for the behavioral sciences*, chapter 4, pages 142–177. 2005.